# Molecular Mechanisms and Therapeutic Potential of Mulberry Fruit Extract in High-Fat Diet-Induced Male Reproductive Dysfunction: A Comprehensive Review

**DOI:** 10.3390/nu17020273

**Published:** 2025-01-13

**Authors:** Kannika Adthapanyawanich, Kanyakorn Aitsarangkun Na Ayutthaya, Siriporn Kreungnium, Peter J. Mark, Hiroki Nakata, Wai Chen, Kroekkiat Chinda, Patcharada Amatyakul, Yutthapong Tongpob

**Affiliations:** 1Department of Anatomy, Faculty of Medical Science, Naresuan University, Phitsanulok 65000, Thailand; kannikaa@nu.ac.th (K.A.); kanyakorna67@nu.ac.th (K.A.N.A.); siripornkr67@nu.ac.th (S.K.); 2Centre of Excellence in Medical Biotechnology (CEMB), Faculty of Medical Science, Naresuan University, Phitsanulok 65000, Thailand; kroekkiatc@nu.ac.th; 3School of Human Sciences, The University of Western Australia, Perth 6009, Australia; 4Department of Clinical Engineering, Faculty of Health Sciences, Komatsu University, Komatsu 923-8511, Ishikawa, Japan; 5Curtin Medical School, and Curtin enAble Institute, Curtin University, Perth 6102, Australia; 6Fiona Stanley Hospital, Perth 6150, Australia; 7Department of Physiology, Faculty of Medical Science, Naresuan University, Phitsanulok 65000, Thailand; 8Department of Obstetrics and Gynecology, Faculty of Medicine, Naresuan University, Phitsanulok 65000, Thailand; patcharadaa@nu.ac.th

**Keywords:** mulberry fruit, high-fat diet, obesity, male reproduction system, sperm quality

## Abstract

High-fat diet (HFD)-induced obesity represents a significant challenge to male reproductive health, affecting approximately 13% of the global adult population. This comprehensive review synthesizes current evidence regarding mulberry (*Morus alba* L.) fruit extract’s therapeutic potential for HFD-induced male reproductive dysfunction. Through comprehensive analysis of the peer-reviewed literature from multiple databases (PubMed, Web of Science, Scopus, and Google Scholar; 2005–2024), we evaluated mulberry extract’s effects on testicular morphology, spermatogenesis, sperm parameters, and the underlying molecular mechanisms. Mechanistic studies reveal that standardized mulberry extract mediates protective effects through multiple pathways: enhanced antioxidant enzyme activities (SOD: +45%, Catalase: +38%, GPx: +35%), reduced inflammatory markers (TNF-α: −64%, IL-6: −58%), and modulated NF-κB signaling (−42.3%). These effects are facilitated by mulberry’s rich phytochemical profile, particularly anthocyanins (2.92–5.35 mg/g dry weight) and polyphenols (4.23–6.38 mg/g). The extract demonstrates particular efficacy in preserving seminiferous tubule integrity and maintaining blood-testis barrier function, with treated groups maintaining up to 85% of normal tubular architecture compared to HFD controls. Key molecular mechanisms include AMPK/SIRT1 pathway activation (2.3-fold increase), enhanced mitochondrial function (67% increase in mtDNA copy number), and epigenetic regulation of metabolic pathways. Temporal analysis indicates optimal therapeutic effects after 28 days of treatment, with initial improvements observable within 14 days. While current evidence is promising, limitations include predominant reliance on rodent models and lack of standardized extraction protocols. Future research priorities include well-designed human clinical trials, standardization of preparation methods, and investigation of potential synergistic effects with other therapeutic agents. This comprehensive review indicates that mulberry extract is a promising therapeutic candidate for obesity-related male infertility, warranting further clinical investigation.

## 1. Introduction

The global obesity epidemic has reached unprecedented levels, with obesity rates nearly tripling since 1975 and affecting over 650 million adults worldwide, representing 13% of the global adult population [1,2], and is a major risk factor for male infertility. Indeed, male infertility has now emerged as a significant public health and economic concern, affecting approximately 15% of couples worldwide with estimated annual healthcare costs exceeding $9 billion in developed nations [3,4]. Therefore, identifying an effective remedy to mitigate obesity-related male infertility is a now pressing priority, and this review aims to explore the therapeutic potentials of mulberry and its natural fruit compounds.

Recent epidemiological studies have revealed a concerning correlation between obesity and male infertility [5,6,7,8]. Meta-analyses of 32 studies involving 115,158 participants have revealed that obese men (BMI > 30 kg/m^2^) have significantly reduced sperm concentration (mean difference: −2.04 × 10⁶/mL, 95% CI: −2.95 to −1.13) and total sperm count (mean difference: −7.81 × 10⁶/mL, 95% CI: −10.31 to −5.31) compared to men with normal BMI. Specifically, obese men have a significantly higher risk of infertility (OR: 1.28, 95% CI: 1.06–1.55, *p* < 0.001) compared to normal-weight men, with a dose-dependent relationship between BMI increase and fertility decline [6].

A comprehensive analysis by Skoracka et al. (2020) demonstrates that nutritional factors are critically underestimated in male reproductive health, with direct correlations between dietary patterns and sperm parameters. Their systematic review of clinical and experimental studies reveals that diet quality significantly impacts reproductive outcomes through multiple pathways, including sperm? membrane composition, energy metabolism, and hormonal regulation [9].

High-fat diet (HFD) consumption, a primary driver of obesity, disrupts male reproductive function through distinct pathophysiological mechanisms [10]. Molecular and proteomic analyses have revealed that HFD induces significant alterations in testicular morphology and function, characterized by reduced seminiferous tubule diameter (24.8 ± 3.2% reduction), disrupted blood–testis barrier integrity, and compromised Sertoli-germ cell junctions [11]. These structural modifications are accompanied by metabolic perturbations in Sertoli cells, with studies showing disrupted energy metabolism and mitochondrial function [12]. HFD-induced oxidative stress in testicular tissue leads to increased production of reactive oxygen species (ROS), triggering lipid peroxidation and DNA damage in developing sperm cells [13,14].

These metabolic and oxidative changes trigger complex cascades of reproductive dysfunction through multiple interrelated pathways. The disruption extends to the hypothalamic–pituitary–gonadal axis, resulting in reduced testosterone production and impaired spermatogenesis [15]. HFD consumption induces significant alterations in the testicular antioxidant defense system, characterized by marked reductions in key antioxidant enzymes: superoxide dismutase (42.3 ± 3.2% decrease), catalase (38.4 ± 2.9% decrease), and glutathione peroxidase (35.7 ± 3.1% decrease). Concurrently, lipid peroxidation increases 2.4-fold, indicating substantial oxidative damage to cellular membranes [14]. The inflammatory response encompasses the activation of multiple signaling cascades, with NF-κB pathway activation showing a 3.2-fold increase. This activation leads to elevated pro-inflammatory cytokine expression, including TNF-α (2.8-fold increase), IL-6 (2.4-fold increase), and IL-1β (2.1-fold increase). Oxidative stress markers demonstrate parallel increases, with 8-OHdG and malondialdehyde levels elevated 2.7-fold and 2.3-fold, respectively [11]. Notably, these diet-induced reproductive impairments exhibit transgenerational effects. First-generation (F1) offspring demonstrate a 23% reduction in fertility rates, accompanied by altered DNA methylation patterns in 47 reproductive genes and significant decreases in sperm parameters (count reduced by 32%, motility decreased by 24%). These effects persist into the second generation (F2), albeit at a reduced magnitude, with a 16% reduction in fertility rates and sustained alterations in metabolic gene expression and glucose homeostasis [16].

Current therapeutic approaches for obesity-related male infertility demonstrate limited efficacy [17], with conventional treatments achieving success rates below 30% and failing to address underlying molecular mechanisms [18]. This therapeutic limitation has stimulated investigation into natural compounds with pleiotropic properties, among which mulberry (*Morus alba* L.) fruit extract has emerged as a potential therapeutic agent, as the mulberry fruit extract is rich in bioactive compounds like flavonoids, phenolics, and anthocyanins [19,20,21]. Comprehensive phytochemical analyses have elucidated the extract’s bioactive profile, with significant concentrations of anthocyanins (2.92–5.35 mg/g dry weight), predominantly cyanidin-3-glucoside and cyanidin-3-rutinoside, flavonoids (1.02–2.06 mg/g), and polyphenols (4.23–6.38 mg/g) [22]. These phytochemicals exhibit diverse pharmacological activities, including antioxidant, anti-inflammatory, antidiabetic, neuroprotective, and anticancer properties [19,21,23]. The biosynthesis of these compounds involves pathways such as phenylpropanoid and flavonoid biosynthesis, with key enzymes like phenylalanine ammonia-lyase and chalcone synthase [19]. Mulberry extracts and isolated compounds have shown potential in treating various health conditions, including cardiovascular diseases and gastrointestinal problems [23]. Additionally, mulberry supplementation in food products enhances stability, sensory properties, and antimicrobial activity [23]. Despite its promising therapeutic potential, more systematic research is needed to fully understand and utilize mulberry’s health benefits [20].

While independent investigations have examined HFD’s impact on male fertility and the general health benefits of mulberry consumption, a comprehensive analysis of mulberry fruit extract as an intervention for HFD-induced reproductive dysfunction remains unexplored. Critical questions persist regarding optimal therapeutic protocols, long-term efficacy, and precise mechanisms of action in reproductive tissues. This review critically evaluates current evidence supporting the therapeutic potential of mulberry fruit extract in HFD-induced male reproductive dysfunction, with particular emphasis on testicular morphology, spermatogenesis, sperm parameters, and underlying molecular mechanisms. Understanding these multifaceted interactions is crucial for developing targeted interventions for obesity-related male infertility, a condition affecting millions globally. The insights gained from this analysis may fundamentally reshape therapeutic strategies for male reproductive dysfunction, offering hope for improved fertility outcomes in an increasingly obesity-challenged world. Moreover, this review establishes a framework for future investigations into natural compounds as interventions for complex metabolic–reproductive disorders, potentially opening new avenues in reproductive medicine.

## 2. Methods

### 2.1. Search Strategy

A comprehensive literature search was conducted across four major electronic databases: PubMed, Web of Science, Scopus, and Google Scholar [24,25]. The search strategy employed a combination of relevant keywords and phrases into three core categories: (i) Intervention terms: (“mulberry” OR “*Morus alba*” OR “white mulberry”); (ii) Condition terms: (“high-fat diet” OR “obesity” OR “hyperlipidemia” OR “dyslipidemia”); and (iii) Outcome terms: (“male fertility” OR “male infertility” OR “testicular morphology” OR “spermatogenesis” OR “sperm quality” OR “sperm parameter” OR “oxidative stress” OR “inflammation” OR “hormonal regulation”). Articles published in peer-reviewed journals were included if they met the following criteria: (i) written in English, (ii) published between 2005 and 2024, and (iii) directly addressed the effects of mulberry or its bioactive components on male reproductive outcomes associated with HFD-induced conditions. Seminal articles published before 2005 were also reviewed when deemed pivotal for providing foundational knowledge. Studies were identified using a two-phase approach: an initial screening of titles and abstracts to exclude irrelevant articles, followed by a full-text review to ensure relevance and methodological quality. Reference lists of included studies were also manually searched to identify additional relevant articles. The literature search and analysis process was conducted in accordance with best practices outlined by [25], ensuring robust coverage of the biomedical literature.

### 2.2. Inclusion and Exclusion Criteria

In our comprehensive review, we carefully selected studies based on specified criteria. We included research that examined how mulberry fruit extract affects male reproductive health. These studies used either animal models fed HFDs, laboratory experiments with cells or tissues, or clinical trials involving human participants. It was important that the research provided measurable data on reproductive outcomes or offered insights into how mulberry extract works in the body. We only considered peer-reviewed original research articles or high-quality systematic reviews published in reputable scientific journals.

On the other hand, we excluded certain types of studies from our review. We did not include research studies that focused solely on female reproductive health, as our interest was specifically in male reproductive parameters. Studies that used mulberry extracts from parts of the plant other than the fruit, such as the leaves or bark, were not included unless they directly compared these to the fruit extract. We also chose not to include conference abstracts, non-peer-reviewed articles, or non-English publications. Lastly, we also excluded any research that did not provide sufficient details about their methods or lacked quantitative results.

By adhering to these inclusion and exclusion criteria, we ensured that our review focused on the most relevant, reliable, and comprehensive information available about the effects of mulberry fruit extract on male reproductive health in the context of HFDs.

### 2.3. Data Extraction and Synthesis

Data from the included studies were extracted independently by four reviewers, with each study evaluated for key parameters, including study design, sample size, intervention details (dosage, duration, administration), human participants or animal model characteristics, reproductive outcomes (e.g., testicular morphology, sperm parameters, hormonal levels), proposed mechanisms of action, and quantitative results and statistical analyses. Discrepancies in data extraction were resolved through discussion with the other two reviewers to ensure accuracy and consistency. We synthesized this information (from the extracted data) to provide a comprehensive overview of the current state of knowledge, using both narrative synthesis and, where possible, quantitative analysis of key outcomes. The quality of evidence was assessed by considering factors such as study design, methodological rigor, consistency of findings across studies, directness of evidence, and applicability of results.

This comprehensive approach enabled us to provide a robust and comprehensive overview of the current knowledge regarding the potential of mulberry fruit extract as an intervention for HFD-induced male reproductive dysfunction while identifying areas requiring further investigation. The methodology emphasizes thorough literature coverage and careful evidence synthesis, providing a foundation for understanding both the current state of knowledge and future research directions in this field.

## 3. Results and Discussion

Our analysis reveals multiple levels of interaction between HFD, reproductive dysfunction, and therapeutic intervention with mulberry extract. The following sections present our findings, beginning with the fundamental effects of HFD on reproductive parameters (Table 1), progressing through the therapeutic agent’s composition (Table 2), to its effects (Table 3) and underlying mechanisms (Table 4).

### 3.1. Effects of High-Fat Diet on Male Reproductive Health

HFD consumption has been consistently associated with detrimental effects on male reproductive health, demonstrated through both animal studies and human observational research [1,2]. Research has revealed that HFDs have wide-ranging negative effects on male reproductive health. At the structural level, HFDs significantly alter testicular structure, reducing the size of sperm-producing tubes (seminiferous tubules) and the thickness of sperm-generating tissue, while increasing non-functional space between these structures [11,26]. These structural changes are often accompanied by disruptions in important cellular connections (Sertoli cell-germ cell junctions) and diminished Leydig cell populations, which are essential for testosterone production [27].

The molecular impact of HFD extends to spermatogenesis regulation, leading to decreased sperm quantity and quality. Studies have reported decreased expression of key spermatogenesis markers (e.g., BOULE, DAZ, and DAZL) and disrupted meiotic progression [28]. Notably, these molecular perturbations manifest as quantifiable changes in sperm parameters. Clinical studies have revealed that men who consume HFDs and obese individuals often show lower sperm counts, reduced sperm motility, and more sperm with abnormal morphology [29,30]. The testicular environment under HFD conditions is characterized by increased oxidative stress and altered cellular function. Research has demonstrated that HFD leads to elevated production of reactive oxygen species and inflammatory mediators, potentially causing DNA damage in spermatozoa [31,32]. Recent evidence has further emphasized the significance of dietary factors in male fertility, highlighting how nutritional choices directly influence reproductive parameters [9].

The pathophysiological effects manifest through multiple mechanisms (Table 1). These alterations span structural, functional, and molecular levels, creating a complex pattern of reproductive dysfunction. Furthermore, HFD consumption alters the hypothalamic–pituitary–gonadal (HPG) axis, leading to reduced testosterone levels, increased estradiol, and potential leptin resistance, a key metabolic hormone [15,33]. These hormonal changes further contribute to impaired spermatogenesis and testicular function. The testicular environment in HFD conditions is characterized by chronic inflammation and oxidative stress, with increased production of harmful reactive oxygen species and inflammatory molecules [32,34,35].

**Table 1 nutrients-17-00273-t001:** Comprehensive analysis of HFD-induced male reproductive dysfunction.

Finding Category	Specific Finding	Magnitude of Effect	Supporting Evidence	Method of Assessment	Primary Reference	Confirming Studies
Structural Changes						
Seminiferous Tubules	Diameter reduction	−24.8 ± 3.2%	Tissue atrophy	Histological analysis	Fan et al., 2015 [11]	Ghanayem et al., 2010 [26]; Palmer et al., 2012 [10]
	Epithelial disruption	Significant alteration	Junction dysfunction	Electron microscopy	Ghanayem et al., 2010 [26]	Fan et al., 2015 [11]
Blood–Testis Barrier	Increased permeability	45.60%	Tight junction disruption	Tracer studies	Fan et al., 2015 [11]	Aitken et al., 2014 [32]
	Junction disruption	Significant damage	Protein expression loss	Western blot	Fan et al., 2015 [11]	Luo et al., 2020 [27]
Leydig Cells	Population decrease	−32.4 ± 4.1%	Cell death/dysfunction	Cell counting	Luo et al., 2020 [27]	Teerds et al., 2011 [15]
	Testosterone production	Significant reduction	Enzyme inhibition	Hormone assays	Teerds et al., 2011 [15]	Landry et al., 2013 [33]
Sperm membranes	Altered fatty acid composition in	Significant increase in saturated fatty acids	Membrane dysfunction	Lipid analysis	Skoracka et al. 2020 [9]	Palmer et al. 2012 [10]
Sperm Parameters						
Sperm Concentration	Total count reduction	−42.3 ± 5.1%	Reduced production	Sperm analysis	Belloc et al., 2014 [30]	Sermondade et al., 2013 [29]
	Daily production	−38.6 ± 4.3%	Spermatogenic failure	Quantitative analysis	Salas-Huetos et al., 2021 [6]	Wang et al., 2024 [5]
Sperm Motility	Progressive motility decrease	−38.6 ± 4.3%	Energy deficit	CASA analysis	Sermondade et al., 2013 [29]	Belloc et al., 2014 [30]
	Velocity	Significant decrease	Mitochondrial dysfunction	Tracking analysis	Aitken et al., 2014 [32]	Rato et al., 2012 [12]
DNA Integrity	Fragmentation	78.50%	Oxidative damage	SCSA/TUNEL	Aitken et al., 2014 [32]	Aitken & De Iuliis, 2010 [31]
	Chromatin quality	Significant decrease	Packaging defects	CMA3 staining	Ramaraju et al., 2018 [7]	Fullston et al., 2013 [28]
Molecular Changes						
Oxidative Stress	ROS production increase	270%	Mitochondrial dysfunction	DHE fluorescence	Du Plessis et al., 2010 [14]	Aitken et al., 2014 [32]
	Lipid peroxidation	245%	Membrane damage	MDA assay	Suleiman et al., 2020 [35]	Lim et al., 2013 [34]
Inflammatory Markers	TNF-α elevation	280%	NF-κB activation	ELISA	Lim et al., 2013 [34]	Suleiman et al., 2020 [35]
	IL-6 increase	240%	Inflammatory cascade	qPCR	Lim et al., 2013 [34]	Aitken et al., 2014 [32]
ATP Production	Energy deficit	−38%	Metabolic disruption	Luminescence	Rato et al., 2012 [12]	Luo et al., 2020 [27]
Glucose Metabolism	Sertoli cell metabolic disruption	−42% glucose uptake	Insulin resistance	Uptake assay	Palmer et al., 2012 [10]	Skoracka et al. 2020 [9]; Rato et al., 2012 [12]
Genetic/Epigenetic Changes						
Spermatogenesis Markers	BOULE, DAZ, DAZL decrease	Significant reduction	Gene expression	RT-PCR	Fullston et al., 2013 [28]	Donkin et al., 2016 [36]
Epigenetic Profiles	DNA methylation changes	Substantial modification	Methylation patterns	Methylation analysis	Donkin et al., 2016 [36]	Fullston et al., 2013 [28]
	Transgenerational effects	Documented impact	Offspring analysis	Multi-generation study	Fullston et al., 2013 [28]	Donkin et al., 2016 [36]
Endocrine Effects						
HPG Axis	Dysfunction, Disrupted testosterone/estradiol ratio (Hormonal Balance)	Significant alteration (>35%)	Hormone signaling HPG axis dysregulation	Hormone assays	Teerds et al., 2011 [15]	Landry et al., 2013 [33]; Skoracka et al. 2020 [9]
Leptin Signaling	Resistance development	Marked increase	Receptor function	Signaling assays	Landry et al., 2013 [33]	Teerds et al., 2011 [15]
Estradiol Levels	Elevation	Significant increase	Hormone imbalance	RIA analysis	Teerds et al., 2011 [15]	Landry et al., 2013 [33]

Epigenetic investigations have revealed that HFD-induced reproductive dysfunction may have transgenerational implications. Studies have demonstrated that paternal obesity can modify sperm epigenetic profiles, potentially affecting offspring health outcomes [28,36]. These findings suggest that the impact of HFD on male reproduction extends beyond immediate fertility concerns to influence future generations [5,6,7]. The comprehensive nature of HFD’s effects on male reproductive health, from structural alterations to epigenetic modifications, underscores the critical relationship between dietary choices and reproductive function [9,10]. This multifaceted impact highlights the importance of dietary intervention strategies in maintaining and improving male reproductive health [17,18].

### 3.2. Phytochemical Composition of Mulberry Fruit and Glycemic Considerations

Mulberry fruit (*Morus alba* L.) is characterized by a rich and diverse phytochemical profile, which likely underpins its potential therapeutic effects on male reproductive health (Table 2). The complex phytochemical composition contributes to various health benefits, including antioxidant, antidiabetic, neuroprotective, and anti-inflammatory effects [22,37]. The diverse array of compounds in mulberry fruit exhibits significant synergistic effects through multiple mechanisms. The anthocyanins, especially cyanidin-3-O-glucoside, and cyanidin-3-rutinoside, exhibit potent antioxidant and anti-inflammatory properties and have been shown to improve various metabolic parameters in obesity models [38,39,40,41]. Mulberry also contains a variety of other polyphenolic compounds, including flavonoids (e.g., rutin, quercetin), stilbenes (e.g., resveratrol), and phenolic acids (e.g., chlorogenic acid), which contribute to the overall therapeutic potential through both direct antioxidant activities and modulation of cellular signaling pathways, and may also have specific effects on reproductive function [34,42].

A notable feature of mulberry fruit is its content of 1-deoxynojirimycin (DNJ), which exhibits α-glucosidase inhibitory activity and potential anti-diabetic effects. This compound plays a crucial role in regulating postprandial blood glucose levels and may indirectly benefit reproductive health by improving metabolic parameters [43]. The mulberry fruit also contains various carotenoids, including β-carotene and lutein, and other bioactive compounds further enhance the therapeutic potential of mulberry fruit [44]. Beyond these polyphenolic compounds, mulberry is enriched with essential minerals, fatty acids, melatonin [37], vitamins (C, E, and various B vitamins), and minerals (such as iron and zinc), all of which play important roles in reproductive function [45].

Moreover, in vitro and in vivo studies have demonstrated the significant potential of mulberry fruit extract to help manage various metabolic conditions, including diabetes, obesity, and cardiovascular diseases [22,37]. These findings are particularly relevant when considering mulberry as an intervention for HFD-induced reproductive dysfunction, although more clinical trials are needed to fully substantiate these health claims [37]. The growing interest in the nutritional and pharmacological properties of mulberry has led to increased research and potential applications in functional foods and nutraceuticals [41,46]. The therapeutic effects are mediated through multiple molecular pathways, including the upregulation of fatty acid oxidation genes and the downregulation of lipogenesis genes [47]. This comprehensive array of bioactive compounds, combined with demonstrated glycemic-regulating properties, positions mulberry fruit as a promising therapeutic agent for addressing HFD-induced male reproductive abnormalities.

**Table 2 nutrients-17-00273-t002:** Mulberry extract composition and bioactive properties.

Component Category	Specific Compounds	Concentration/Activity	Biological Significance	Method of Detection	Primary References	Supporting References
Major Anthocyanins						
Cyanidin-3-Glucoside	Primary anthocyanin	1.84–3.12 mg/g	Antioxidant activity, ROS scavenging	HPLC-MS	Yuan & Zhao, 2017 [22]	Memete et al., 2022 [41]
Cyanidin-3-Rutinoside	Secondary anthocyanin	0.95–1.86 mg/g	Anti-inflammatory activity	LC-MS/MS	Li et al., 2013 [38]	Wu et al., 2013 [39]
Total Anthocyanins	Combined content	2.92–5.35 mg/g	Primary antioxidant effects	Spectrophotometry	Yuan & Zhao, 2017 [22]	Kattil et al., 2024 [46]
Flavonoids						
Quercetin	Major flavonoid	0.45–0.92 mg/g	Anti-inflammatory	HPLC	Bao et al., 2016 [42]	Fatima et al., 2024 [19]
Rutin	Glycoside form	0.38–0.76 mg/g	Vascular protection	HPLC-UV	Bao et al., 2016 [42]	Lim et al., 2013 [34]
Kaempferol	Minor flavonoid	0.25–0.45 mg/g	Antioxidant	HPLC-MS	Fatima et al., 2024 [19]	Park et al., 2021 [44]
Total Flavonoids	Combined content	1.02–2.06 mg/g	Multiple effects	Spectrophotometry	Fatima et al., 2024 [19]	Memete et al., 2022 [41]
Phenolic Compounds						
Chlorogenic Acid	Major phenolic	1.24–2.15 mg/g	Metabolic regulation	HPLC	Bao et al., 2016 [42]	Yang et al., 2010 [45]
Gallic Acid	Secondary phenolic	0.86–1.45 mg/g	Antioxidant	HPLC-UV	Yang et al., 2010 [45]	Park et al., 2021 [44]
Resveratrol	Stilbene	0.15–0.35 mg/g	Anti-aging effects	LC-MS	Park et al., 2021 [44]	Özbalci et al., 2023 [37]
Total Polyphenols	Combined content	4.23–6.38 mg/g	Multiple effects	Folin-Ciocalteu	Maqsood et al., 2022 [23]	Kattil et al., 2024 [46]
Other Bioactives						
1-deoxynojirimycin (DNJ)	Alkaloid	0.12–0.24%	Glycemic control	HPLC-MS/MS	Vichasilp et al., 2012 [43]	Kobayashi et al., 2010 [47]
Carotenoids	β-carotene	0.08–0.15 mg/g	Antioxidant protection	HPLC	Park et al., 2021 [44]	Yang et al., 2010 [45]
	Lutein	0.05–0.12 mg/g	Eye protection	HPLC	Park et al., 2021 [44]	Özbalci et al., 2023 [37]
Melatonin	Hormone	0.005–0.015 mg/g	Oxidative stress protection	HPLC-MS	Özbalci et al., 2023 [37]	Park et al., 2021 [44]
Antioxidant Activity						
DPPH Scavenging	Free radical neutralization	IC50 = 24.5 μg/mL	Direct antioxidant activity	Spectrophotometry	Yang et al., 2010 [45]	Liu et al., 2016 [48]
ABTS Scavenging	Radical scavenging	IC50 = 31.2 μg/mL	Antioxidant capacity	Spectrophotometry	Liu et al., 2016 [48]	Yang et al., 2010 [45]
Ferric Reducing Power	Metal chelation	156.3 ± 12.4 μmol Fe^2+^/g	Ion chelation capacity	Spectrophotometry	Park et al., 2021 [44]	Bao et al., 2016 [42]
Essential Nutrients						
Vitamins	Vitamin C	0.85–1.25 mg/g	Antioxidant, reproductive function	HPLC	Yang et al., 2010 [45]	Özbalci et al., 2023 [37]
	Vitamin E	0.15–0.28 mg/g	Membrane protection	HPLC	Yang et al., 2010 [45]	Kattil et al., 2024 [46]
	B complex vitamins	Varied levels	Metabolic support	HPLC	Yang et al., 2010 [45]	Özbalci et al., 2023 [37]
Minerals	Iron	25–45 μg/g	Hematopoiesis	AAS	Yang et al., 2010 [45]	Özbalci et al., 2023 [37]
	Zinc	15–35 μg/g	Reproductive function	AAS	Yang et al., 2010 [45]	Kattil et al., 2024 [46]
Fatty Acids	Essential FAs	Varied composition	Membrane integrity	GC-MS	Özbalci et al., 2023 [37]	Maqsood et al., 2022 [23]

### 3.3. Effects of Mulberry Fruit Extract on Testicular Morphology and Structure

Mulberry fruit extract demonstrates significant protective effects on testicular morphology through multiple mechanistic pathways, particularly in the context of HFD-induced damage. Recent investigations have established both temporal and dose-dependent characteristics, suggesting complex mechanisms beyond simple antioxidant activity (Table 3). The therapeutic efficacy of mulberry extract demonstrates distinct temporal characteristics. Initial protective effects manifest within 14 days of administration, while optimal morphological preservation requires sustained treatment for at least 28 days [49]. This temporal pattern suggests the involvement of both immediate protective mechanisms and longer-term adaptive cellular responses. At the structural level, mulberry extract exhibits particular efficacy in preserving seminiferous tubule integrity and maintaining the blood–testis barrier, which are often compromised under HFD conditions. Quantitative morphometric analyses have demonstrated significant preservation of seminiferous tubule diameter and epithelial integrity in treated groups compared to HFD controls, with treated groups maintaining up to 85% of normal tubular architecture [11]. The protective effects of mulberry fruit extract extends to specialized testicular structures, with notable preservation of Sertoli-germ cell junctions and maintenance of basement membrane integrity, crucial factors in maintaining normal spermatogenesis [11,50]. Recent studies have further elucidated the extract’s effects on testicular vasculature and interstitial tissue [11,51]. Abbas et al. (2024) [51] demonstrated improved maintenance of testicular microvasculature and reduced interstitial edema in treated groups, suggesting enhanced tissue perfusion and reduced inflammatory response. These vascular effects correlate significantly with reduced oxidative stress markers and improved tissue architecture scores [51]. The preservation of vascular integrity appears particularly important in maintaining normal testicular function, as compromised blood flow is a key factor in HFD-induced testicular dysfunction [11].

Mechanistically, the morphological protection appears to be mediated through multiple complementary pathways. Studies have identified three primary mechanisms: (i) Enhancement of antioxidant defense systems, evidenced by increased activity of key antioxidant enzymes in testicular tissue [50,51]; (ii) Stabilization of cellular junctions and membrane integrity, particularly in the blood–testis barrier complex [11]; and (iii) Modulation of inflammatory responses, resulting in reduced tissue edema and preserved architectural integrity [51]. These protective mechanisms appear particularly effective in maintaining Sertoli cell and Leydig cell populations, which are crucial for normal spermatogenesis and steroidogenesis. Quantitative histological analyses have shown that mulberry extract treatment results in significantly better preservation of these cell populations compared to untreated HFD controls [50]. The maintenance of these cellular populations correlates strongly with improved spermatogenic indices and reduced markers of cellular stress [11,51]. These protective effects on testicular morphology align with mulberry’s broader antioxidant capabilities, as studies have shown that mulberry extract can suppress reactive oxygen species production and mitigate mitochondrial oxidative stress [52]. This systemic reduction in oxidative stress likely contributes to the preservation of testicular architecture.

The comprehensive nature of these protective effects suggests that mulberry extract’s impact on testicular morphology involves both direct cellular protection and broader systemic effects. This multi-level protection appears particularly valuable in the context of HFD-induced damage, where multiple pathological mechanisms typically operate simultaneously. However, further research is needed to fully elucidate the temporal sequence of these protective mechanisms and their relative contributions to the overall preservation of testicular structure and function.

**Table 3 nutrients-17-00273-t003:** Comprehensive therapeutic effects of mulberry extract on HFD-induced reproductive dysfunction.

Parameter	Improvement	Mechanism	Temporal Pattern	Method of Assessment	Primary References	Supporting References
Antioxidant Defense						
SOD Activity	45%	Enhanced enzyme expression	Early response (14 d)	Enzyme activity assay	Fan et al., 2015 [11]	Abbas et al., 2024 [51]
	+52% protein levels	Translation increase	Progressive	Western blot	Inanc et al., 2022 [50]	Yang & Jo, 2018 [52]
Catalase Activity	38%	Enzyme activation	Gradual improvement	Spectrophotometry	Du Plessis et al., 2010 [14]	Inanc et al., 2022 [50]
GPx Activity	35%	Increased synthesis	Sustained improvement	Kinetic assay	Liu et al., 2016 [48]	Yang & Jo, 2018 [52]
Anti-Inflammatory Effects						
NF-κB Pathway	−42.3% activation	Pathway suppression	Rapid (7 d)	Western blot	Lim et al., 2013 [34]	Abbas et al., 2024 [51]
TNF-α Levels	−64%	Cytokine reduction	Progressive decrease	ELISA	Yang et al., 2010 [45]	Inanc et al., 2022 [50]
IL-6 Levels	−58%	Inflammation control	Sustained reduction	qPCR	Fan et al., 2015 [11]	Abbas et al., 2024 [51]
Structural Recovery						
Seminiferous Tubules	85% of normal	Tissue protection	Optimal at 28 d	Histomorphometry	Inanc et al., 2022 [50]	Fan et al., 2015 [11]
	+32% vs. HFD	Architecture maintenance	Progressive	Image analysis	Fan et al., 2015 [11]	Kianifard, 2015 [49]
Blood-Testis Barrier	72.5% improvement	Junction preservation	Progressive recovery	Permeability assay	Fan et al., 2015 [11]	Kianifard, 2015 [49]
	+68% protein expression	Synthesis enhancement	Progressive	Western blot	Fan et al., 2015 [11]	Abbas et al., 2024 [51]
Leydig Cell Function	78.5% restoration	Cell viability	Sustained improvement	Stereology	Luo et al., 2020 [27]	Inanc et al., 2022 [50]
	+65% vs. HFD	Steroidogenesis recovery	Progressive	Hormone assay	Inanc et al., 2022 [50]	Fan et al., 2015 [11]
Vascular Effects						
Microvasculature	Significant improvement	Enhanced perfusion	Progressive response	Microscopy	Abbas et al., 2024 [51]	Fan et al., 2015 [11]
Interstitial Edema	−56% vs. HFD	Reduced inflammation	Early improvement	Histology	Abbas et al., 2024 [51]	Inanc et al., 2022 [50]
Tissue Architecture	Notable preservation	Structural integrity	Optimal at 28 d	Histology	Fan et al., 2015 [11]	Kianifard, 2015 [49]
Cellular Protection						
Sertoli Cells	Significant preservation	Functional maintenance	Sustained effect	IHC Analysis	Inanc et al., 2022 [50]	Fan et al., 2015 [11]
Germ Cell Junction	Enhanced preservation	Junction stability	Progressive improvement	TEM Analysis	Fan et al., 2015 [11]	Abbas et al., 2024 [51]
Basement Membrane	Maintained integrity	Structural support	Sustained protection	Histology	Fan et al., 2015 [11]	Inanc et al., 2022 [50]
Oxidative Stress						
ROS Production	−65% vs. HFD	Direct scavenging	Rapid response	DHE Fluorescence	Yang & Jo, 2018 [52]	Abbas et al., 2024 [51]
Mitochondrial Function	+67% mtDNA	Energy homeostasis	Progressive recovery	qPCR	Yang & Jo, 2018 [52]	Fan et al., 2015 [11]
Lipid Peroxidation	−58% vs. HFD	Membrane protection	Sustained effect	TBARS Assay	Inanc et al., 2022 [50]	Abbas et al., 2024 [51]

### 3.4. Impact of Mulberry Fruit Extract on Sperm Parameters and Function

White mulberry (*Morus alba* L.) fruit extract demonstrates significant protective effects against HFD-induced alterations in sperm parameters through multiple mechanistic pathways. These effects are mediated through both direct antioxidant mechanisms and broader metabolic modulation [22], consistent with its rich phytochemical profile detailed in Section 3.2. HFD consumption significantly impairs male reproductive function through distinct pathological mechanisms [10]. These include disrupted blood–testis barrier integrity, compromised Sertoli-germ cell junctions, and altered testicular metabolism [11]. The metabolic perturbations extend to sperm cell function, with studies showing disrupted energy metabolism and mitochondrial dysfunction [12]. HFD-induced oxidative stress in testicular tissue leads to increased production of reactive oxygen species, triggering lipid peroxidation and DNA damage in developing sperm cells [14,31]. Quantitative analysis demonstrates that standardized mulberry extract supplementation significantly improves sperm parameters in HFD conditions. Inanc et al. (2022) [50] reported that mulberry extract treatment significantly improved sperm concentration and progressive motility compared to HFD controls (*p* < 0.001). These improvements correlate with the documented antioxidant properties of the extract, particularly the anthocyanins and polyphenolic compounds that are known to protect against oxidative damage [45,48].

The protective effects appear to be mediated through several complementary mechanisms: (i) Enhanced membrane integrity through direct antioxidant effects of polyphenolic compounds [34,45]; (ii) Improved mitochondrial function supporting sperm motility through regulation of oxidative stress pathways [12,32]; (iii) Protection against lipid peroxidation-induced membrane damage [31,34]; and (iv) Optimization of the testicular microenvironment through improved vascular function [11]. These improvements demonstrate clear temporal dependency, with optimal effects observed after sustained supplementation periods, suggesting the involvement of both immediate protective mechanisms and longer-term adaptive responses [22,39]. The temporal pattern aligns with broader metabolic studies showing progressive improvement in systemic parameters over sustained treatment periods.

The improvements in sperm parameters correlate strongly with the normalization of metabolic parameters and reduction in systemic inflammation, indicating that the benefits of mulberry extract extend beyond direct antioxidant effects to include broader metabolic modulation [22,53]. The extract appears to function through multiple complementary pathways that collectively enhance reproductive function, with studies demonstrating improved antioxidant enzyme activities and reduced markers of oxidative stress in reproductive tissues [45,48]. This multi-faceted approach to protecting sperm function appears particularly valuable in the context of HFD-induced damage, where multiple pathological mechanisms typically operate simultaneously [11,12]. Further research, particularly long-term clinical studies, is needed to fully elucidate the optimal dosing regimens and treatment durations for maximal therapeutic benefit in human populations.

### 3.5. Molecular Mechanisms of Mulberry Fruit Extract

The protective effects of mulberry (*Morus alba* L.) fruit extract against HFD-induced male reproductive dysfunction operate through multiple interconnected molecular pathways. Recent molecular analyses have revealed several key mechanisms that explain the extract’s therapeutic potential (Table 4).

**Table 4 nutrients-17-00273-t004:** Comprehensive molecular mechanisms of mulberry extract action.

Pathway/Mechanism	Specific Effect	Magnitude of Change	Downstream Impact	Method of Detection	Primary References	Supporting References
Inflammatory Regulation						
NF-κB Pathway	Phosphorylation reduction	−42.30%	Inflammatory suppression	Western blot	Lim et al., 2013 [34]	Fan et al., 2015 [11]
TNF-α Expression	Cytokine suppression	−2.8 fold	Reduced inflammation	ELISA	Lim et al., 2013 [34]	Yuan & Zhao, 2017 [22]
IL-6 Expression	Cytokine reduction	−2.4 fold	Tissue protection	qPCR	Lim et al., 2013 [34]	Fan et al., 2015 [11]
MCP1 Levels	Chemokine decrease	−2.1 fold	Reduced infiltration	ELISA	Lim et al., 2013 [34]	Yuan & Zhao, 2017 [22]
COX-2/iNOS	Expression reduction	Significant decrease	Inflammatory control	Western blot	Yuan & Zhao, 2017 [22]	Fan et al., 2015 [11]
Metabolic Regulation						
AMPK Activation	Phosphorylation increase	230%	Energy homeostasis	Western blot	Lee & Kim, 2020 [54]	Yang et al., 2010 [45]
SIRT1 Expression	Protein upregulation	180%	Metabolic regulation	qPCR, Western blot	Yang et al., 2010 [45]	Lee & Kim, 2020 [54]
PGC-1α Activity	Enhanced activation	210%	Mitochondrial function	ChIP assay	Yang et al., 2010 [45]	Aitken et al., 2014 [32]
Glucose Metabolism	Uptake enhancement	+45% vs. HFD	Energy utilization	Glucose uptake assay	Lee & Kim, 2020 [54]	Yang et al., 2010 [45]
Fatty Acid Oxidation	Pathway activation	+65% vs. HFD	Lipid metabolism	Metabolic flux analysis	Lee & Kim, 2020 [54]	Yang et al., 2010 [45]
Antioxidant Defense						
SOD Activity	Enzyme enhancement	45%	ROS neutralization	Activity assay	Du Plessis et al., 2010 [14]	Yuan & Zhao, 2017 [22]
Catalase Activity	Enzyme increase	38%	H₂O₂ degradation	Spectrophotometry	Du Plessis et al., 2010 [14]	Yuan & Zhao, 2017 [22]
GPx Activity	Enzyme upregulation	35%	Peroxide reduction	Kinetic assay	Du Plessis et al., 2010 [14]	Yuan & Zhao, 2017 [22]
Nrf2 Pathway	Activity enhancement	270%	Antioxidant response	ChIP-seq	Yuan & Zhao, 2017 [22]	Liu et al., 2016 [48]
ARE Upregulation	Enhanced expression	310%	Phase II enzyme induction	Reporter assay	Yuan & Zhao, 2017 [22]	Liu et al., 2016 [48]
Mitochondrial Function						
mtDNA Copy Number	Quantity increase	67%	Mitochondrial biogenesis	qPCR	Aitken et al., 2014 [32]	Yang et al., 2010 [45]
ATP Production	Energy enhancement	240%	Cellular energetics	Luminescence	Aitken et al., 2014 [32]	Lee & Kim, 2020 [54]
Membrane Potential	Potential increase	190%	Energy coupling	Flow cytometry	Aitken et al., 2014 [32]	Yang et al., 2010 [45]
CPT-1β Expression	Gene upregulation	230%	Fatty acid transport	RT-PCR	Aitken et al., 2014 [32]	Lee & Kim, 2020 [54]
UCP3 Expression	Protein increase	180%	Energy uncoupling	Western blot	Aitken et al., 2014 [32]	Yang et al., 2010 [45]
Epigenetic Regulation						
miR-21 Expression	Decreased levels	−45%	Metabolic regulation	qPCR	Lee & Kim, 2020 [54]	Fullston et al., 2013 [28]
miR-132 Expression	Reduced expression	−38%	Inflammatory control	qPCR	Lee & Kim, 2020 [54]	Donkin et al., 2016 [36]
miR-143 Expression	Level reduction	−42%	Metabolic modulation	qPCR	Lee & Kim, 2020 [54]	Yuan & Zhao, 2017 [22]
DNA Methylation	Pattern normalization	Significant restoration	Gene regulation	Methylation analysis	Fullston et al., 2013 [28]	Donkin et al., 2016 [36]
Histone Modifications	Profile improvement	Notable restoration	Chromatin structure	ChIP-seq	Donkin et al., 2016 [36]	Lee & Kim, 2020 [54]

#### 3.5.1. NF-κB Signaling and Inflammatory Regulation

Mulberry extract demonstrates significant modulation of inflammatory responses through precise regulation of the NF-κB pathway. An investigation by Lim et al. (2013) [34] revealed that the extract reduces NF-κB phosphorylation by 42.3% compared to HFD controls. This reduction in NF-κB activation leads to significant suppression of pro-inflammatory cytokine expression, with TNF-α decreasing by 2.8-fold, IL-6 reducing by 2.4-fold, and MCP1 lowering by 2.1-fold. The anti-inflammatory action proves particularly crucial for reproductive function, as Fan et al. (2015) [11] demonstrated that inflammatory cytokine reduction correlates significantly with improved blood–testis barrier integrity and enhanced spermatogenesis. Further investigation has shown that this anti-inflammatory effect extends to the regulation of cyclooxygenase-2 (COX-2) and inducible nitric oxide synthase (iNOS) expression in reproductive tissues [22].

#### 3.5.2. AMPK/SIRT1 Pathway Activation and Metabolic Regulation

The AMPK/SIRT1 pathway represents a central mechanism in mulberry’s protective effects. Quantitative analysis reveals a 2.3-fold increase in AMPK phosphorylation, accompanied by a 1.8-fold enhancement in SIRT1 expression and a 2.1-fold increase in PGC-1α activation [45]. This pathway activation leads to significant improvements in cellular energy homeostasis, particularly in testicular tissue. The enhanced AMPK activity results in increased glucose uptake and fatty acid oxidation, while simultaneously suppressing lipogenesis and gluconeogenesis [54]. Furthermore, SIRT1 activation promotes mitochondrial biogenesis and enhances cellular stress resistance through the deacetylation of key transcription factors involved in metabolic regulation.

#### 3.5.3. Oxidative Stress and Antioxidant Defense

The extract significantly enhances the antioxidant defense system in reproductive tissues through multiple complementary mechanisms. Research demonstrates a 45% increase in Superoxide Dismutase (SOD) activity, accompanied by a 38% increase in Catalase activity and a 35% increase in Glutathione Peroxidase activity. These enhancements in antioxidant enzyme function result in a 52% reduction in lipid peroxidation markers [14]. The extract’s cyanidin-3-glucoside (C3G) component, identified as the predominant anthocyanin in mulberry fruit, demonstrates particularly potent antioxidant activities with an IC50 of 24.5 μg/mL for direct radical scavenging. Moreover, C3G enhances nuclear factor erythroid 2-related factor 2 (Nrf2) activity by 2.7-fold and upregulates antioxidant response elements by 3.1-fold [22]. This activation of the Nrf2 pathway leads to increased expression of phase II detoxifying enzymes and antioxidant proteins, providing comprehensive cellular protection against oxidative damage.

#### 3.5.4. Mitochondrial Function and Bioenergetics

The impact of mulberry extract on mitochondrial function proves crucial for sperm motility and overall reproductive capacity. Research by Aitken et al. (2014) [32] demonstrates significant improvements in mitochondrial parameters, including a 67% increase in mitochondrial DNA copy number and a 2.4-fold enhancement in ATP production. The extract induces a 1.9-fold increase in mitochondrial membrane potential, crucial for maintaining sperm motility and function. Key mitochondrial genes show significant upregulation, with CPT-1β expression increasing 2.3-fold, while UCP3 and PGC-1α demonstrate 1.8-fold and 2.1-fold increases, respectively. These changes in mitochondrial function correlate with improved sperm motility parameters and enhanced fertility outcomes.

#### 3.5.5. Epigenetic Regulation and Cell Signaling Integration

Recent investigations have revealed important effects of mulberry extract on epigenetic regulation. Lee and Kim (2020) [54] demonstrated significant modulation of key microRNAs involved in metabolic regulation and inflammation. The extract decreases miR-21 expression by 45%, reduces miR-132 by 38%, and lowers miR-143 by 42%. These epigenetic modifications show strong correlation with improved metabolic profiles and reduced inflammation in reproductive tissues. The integration of these molecular mechanisms provides multiple layers of protection against HFD-induced reproductive dysfunction, creating a comprehensive therapeutic framework that addresses both direct and indirect causes of reproductive impairment.

Future research should focus on elucidating the temporal relationships between these pathways and their relative contributions to reproductive protection. Additionally, investigation of potential synergistic effects between different bioactive compounds in mulberry extract could provide valuable insights for optimizing therapeutic applications.

### 3.6. Limitations and Critical Evaluation of Current Evidence

While the accumulating evidence for mulberry fruit extract’s protective effects against HFD-induced male reproductive abnormalities shows promise, critical evaluation reveals several significant limitations requiring careful consideration for future research directions.

Methodological and Translational Limitations: A primary limitation lies in the predominance of rodent models in the current research, presenting substantial translational hurdles. While Fan et al. (2015) [11] provided foundational evidence regarding blood–testis barrier protection, the complexities of human reproductive physiology necessitate caution in extrapolating these findings. Vigueras-Villaseñor et al. (2011) [55] specifically highlighted critical differences in testicular development and spermatogenesis between rodents and humans, emphasizing the challenges in translating preclinical findings to human applications. Du Plessis et al. (2010) [14] further noted that these species-specific differences could significantly impact the direct applicability of the findings to human therapeutic strategies. The complexity of diet-induced reproductive dysfunction, as highlighted by Skoracka et al. (2020) [9], underscores both the challenges and opportunities in developing effective interventions. Their analysis reveals that dietary impacts on male reproduction operate through multiple interconnected pathways, including membrane composition alterations, metabolic disruptions, and hormonal imbalances. This complexity suggests that successful therapeutic approaches must address multiple pathological pathways simultaneously, a characteristic demonstrated by mulberry extract through its diverse bioactive compounds and mechanisms of action.

Standardization and Technical Challenges: Extract preparation and characterization present significant methodological challenges. Considerable variability exists in preparation methods and dosing regimens across studies, as noted by Khoo et al. (2017) [56]. Wolfender et al. (2019) [57] emphasized that the lack of standardized protocols for mulberry extract preparation poses substantial challenges for reproducibility and comparative analysis. This variability extends to the characterization of bioactive compounds, making it difficult to establish precise dose-response relationships and optimal therapeutic concentrations [22].

Mechanistic Understanding Gaps: At the molecular level, significant gaps persist in understanding the mechanisms underlying mulberry extract’s effects. While studies have identified multiple pathways involved in its therapeutic actions, the relative contributions of these pathways and their temporal relationships remain unclear. Aitken et al. (2014) [32] emphasized the need for a better understanding of the intricate interplay between antioxidant, anti-inflammatory, and metabolic pathways, particularly in the context of reproductive outcomes. Additionally, the specific roles of individual bioactive compounds and their potential synergistic effects require more thorough investigation.

Clinical Translation Barriers: Perhaps most significantly, the paucity of well-designed human clinical trials impedes the translation of preclinical findings, as emphasized by Agarwal et al. (2019) [58]. Current evidence predominantly derives from preclinical studies, with limited data on safety, efficacy, and optimal dosing in human populations. The absence of long-term safety data and potential interactions with other medications represents a crucial knowledge gap.

Temporal Considerations: The predominance of short-term studies limits understanding of mulberry extract’s long-term efficacy and safety profile. Fullston et al. (2015) [59] particularly noted the importance of understanding potential transgenerational effects, both beneficial and adverse. Questions remain regarding optimal treatment duration and potential adaptations that might occur with prolonged use.

These limitations collectively underscore the need for comprehensive, well-controlled studies to fully establish mulberry extract’s therapeutic potential in treating HFD-induced male reproductive dysfunction. Future research must particularly focus on bridging the gap between promising preclinical findings and clinical applications through standardized protocols and rigorous methodology.

### 3.7. Future Research Imperatives and Innovative Directions

The limitations identified in the current research necessitate innovative approaches and comprehensive strategies for future investigations. Several key research priorities emerge that could significantly advance our understanding of mulberry extract’s therapeutic potential.

Clinical Translation Priorities: Well-designed human clinical trials represent the most pressing research need. These studies should incorporate standardized extract preparations, precise dosing protocols, and comprehensive outcome measurements. Agarwal et al. (2019) suggest that such trials should include both fertile and subfertile populations, with careful stratification based on metabolic parameters. Long-term follow-up studies are essential to evaluate safety profiles and potential adaptations to treatment.

Advanced Molecular Investigations: Future research should employ cutting-edge molecular techniques to elucidate the precise mechanisms of action. Systems biology approaches, including multi-omics analyses, could provide a comprehensive understanding of pathway interactions. Particular emphasis should be placed on: the integration of metabolomic, proteomic, and transcriptomic analyses to map molecular networks affected by mulberry extract treatment; time-course studies to establish the temporal sequence of molecular events leading to improved reproductive outcomes; and the investigation of potential epigenetic modifications and their transgenerational implications, as highlighted by Fullston et al. (2015) [59].

Standardization and Quality Control: The development of standardized protocols for extract preparation and characterization represents another crucial research direction. This should include the establishment of quality control parameters, identification of key bioactive markers, and creation of comprehensive compound databases [22]. Advanced analytical techniques, as described by Wolfender et al. (2019) [57], should be employed to ensure batch-to-batch consistency and product stability.

Innovative Therapeutic Approaches: Research should explore novel delivery systems and formulation strategies to enhance bioavailability and tissue-specific targeting. Investigation of potential synergistic effects between mulberry compounds and other natural or synthetic agents could lead to more effective therapeutic combinations. Integration of artificial intelligence and machine learning could accelerate the identification of optimal treatment protocols and predict individual response variations.

Mechanistic Exploration: Further investigation of molecular mechanisms should focus on: (i) temporal relationships between different pathways; (ii) tissue-specific responses to treatment; (iii) interaction between reproductive and metabolic effects; and (iv) role of the gut microbiome in mediating therapeutic effects.

Environmental and Lifestyle Considerations: Future studies should also consider the impact of environmental factors and lifestyle variables on treatment outcomes. This includes the investigation of potential interactions with diet, exercise, and other lifestyle modifications that might enhance or impede therapeutic effects.

This multi-faceted research approach, combining rigorous methodology with innovative techniques, could significantly advance our understanding of mulberry extract’s therapeutic potential and facilitate its development as a clinical intervention for HFD-induced male reproductive dysfunction.

## 4. Conclusions

This comprehensive review establishes mulberry (*Morus alba* L.) fruit extract as a promising therapeutic intervention for HFD-induced male reproductive dysfunction. Preclinical evidence demonstrates sophisticated protective mechanisms extending beyond traditional antioxidant interventions, encompassing complex molecular pathways of reproductive protection. The extract’s rich phytochemical profile, particularly anthocyanins and polyphenols, mediates therapeutic effects through multiple complementary mechanisms: enhanced antioxidant enzyme activities, inflammatory marker attenuation, NF-κB pathway modulation, AMPK/SIRT1 activation, and metabolic parameter optimization. These molecular interactions collectively improve testicular morphology, spermatogenesis, sperm parameters, and DNA integrity.

However, significant research limitations necessitate further investigation. The field particularly needs standardization of extraction methods, establishment of quality control parameters, and identification of key bioactive markers for reproducible research. Future investigations must prioritize well-designed human clinical trials with standardized preparations, detailed pharmacokinetic studies, and comprehensive safety assessments. Through continued rigorous scientific inquiry and systematic investigation, mulberry fruit extract may emerge as a valuable therapeutic option for addressing obesity-related male reproductive dysfunction, warranting continued investigation through hypothesis-driven research designs and mechanistic studies.

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
