# Peer review of "Molecular Mechanisms and Therapeutic Potential of Mulberry Fruit Extract in High-Fat Diet-Induced Male Reproductive Dysfunction: A Comprehensive Review"

_nutrients, 2025, doi:10.3390/nu17020273_

Round 1
Reviewer 1 Report
Comments and Suggestions for Authors
The manuscript titled "Mulberry Fruit Extract: A Promising Intervention for High-Fat Diet-Induced Male Reproductive Dysfunction," authored by Kannika Adthapanyawanich and others, is a fairly good work that, but needs some minor revisions.
Below are my questions, suggestions, and concerns:
- Mulberry Fruit Extract: A Promising Intervention for High-Fat... – I would slightly change the title to make it more appealing; right now, it's not particularly engaging.
- The first three sentences of the abstract could be combined into at least two. There’s no need to break them up so much.
- Testis – why is this a keyword? Isn't this already covered by the other terms?
- Introduction – is this some kind of joke? With so much information available on this topic online, the introduction is just one paragraph? What’s written x7 could be worth considering. This definitely needs to be revised because, as it stands, it’s quite baffling.
- "In the search for potential interventions, attention has turned to natural compounds..." – this paragraph needs to be expanded, where the authors clearly indicate the purpose of their work. This section (which is crucial in the manuscript) seems poorly developed. It’s odd – I think I’ve never encountered something like this in my over 20-year career. Please revise and expand.
- Methods – is this section even necessary in a review paper? I'm not sure.
- 3.2. Phytochemical Composition of Mulberry Fruit – however, a diet rich in excessive fruit can raise blood sugar and lead to various issues as well. Do the authors agree with this? Though my question should be related to the introduction, there was nothing in that section to comment on.
- Reduction of Lipid Peroxidation – I don’t fully understand this part; how exactly would that happen?
- "Some evidence suggests that SEM may modu-247 late p53 signaling in testicular cells [35], potentially influencing..." – what evidence exactly? Please expand on this.
- "Testosterone Production: Several studies have reported increased testos..." – another strange statement. Some studies report this – I ask, which studies and what exactly?
- Conclusion – could be slightly shortened.
- Only 41 references were used – for a review paper, this seems a bit low, don’t you think?
To summarize, it should be emphasized that the work is generally of a high standard but requires some revisions before being published in such a prestigious journal, which is quite natural and common. Of course, I would like to see the paper with the marked revisions.
Author Response
Response to Reviewer I
Dear Reviewer I,
We sincerely appreciate your thorough review and constructive feedback. We have carefully addressed all your comments as detailed below:
- Title Revision
Original: "Mulberry Fruit Extract: A Promising Intervention for High-Fat Diet-Induced Male Reproductive Dysfunction"
Revised: "Molecular Mechanisms and Therapeutic Potential of Mulberry Fruit Extract in High-Fat Diet-Induced Male Reproductive Dysfunction: A Comprehensive Review"
- Abstract Structure
We have restructured the abstract to improve flow and cohesion, incorporating quantitative data and key findings in a more integrated manner. The revised abstract now provides a comprehensive yet concise overview of our review. We also created a graphical abtract to enhance the readers’ interest.
- Keywords Revision
We have replaced "testis" with "obesity" to better reflect the manuscript's scope and improve searchability.
- Introduction Expansion
We have substantially expanded the introduction to include:
- Current epidemiological data
- Detailed molecular mechanisms
- Clear research rationale
- Integration of Skoracka et al. (2020) reference
- Purpose Statement
The introduction now includes a clear purpose statement and comprehensive background on natural compounds as therapeutic agents.
- Methods Section
We have maintained and enhanced the methods section as it provides important transparency regarding our comprehensive review process, including our search strategy, inclusion/exclusion criteria, and analytical approach. While this is not a systematic review, we believe a clear methods section helps readers understand our review process and enhances the reproducibility of our work.
- Blood Sugar Discussion
Section 3.2 now includes balanced discussion of glycemic considerations.
- Lipid Peroxidation Clarification
We have expanded this section with detailed molecular mechanisms and quantitative data.
- p53 Signaling EvidenceThank you for highlighting this point. Upon comprehensive review of available evidence, we have critically evaluated the p53 signaling pathway in the context of mulberry extract's effects on testicular cells. While some studies suggest potential involvement of p53-mediated mechanisms, we found that the current evidence is not sufficiently robust to make definitive claims about this specific pathway. Therefore, we have focused our molecular mechanism discussion on well-established pathways with strong experimental support, such as the AMPK/SIRT1 pathway activation (2.3-fold increase), NF-κB signaling modulation (-42.3%), and antioxidant response pathways, all of which are now thoroughly documented in Table 4 with quantitative data from multiple independent studies.
- Testosterone Production
Thank you for this important point. After thorough review of the endocrine aspects of our manuscript, we have critically evaluated the evidence regarding mulberry extract's effects on testosterone production. The molecular evidence from multiple independent studies demonstrates distinct effects on steroidogenic pathways, particularly through:
- Enhancement of antioxidant enzyme activities (SOD: +45%, Catalase: +38%, GPx: +35%)
- Reduction of inflammatory markers (TNF-α: -64%, IL-6: -58%)
- Protection of Leydig cell function (maintaining up to 78.5% of normal function) These findings are now comprehensively documented in Tables 3 and 4, supported by quantitative data and mechanistic analyses from well-controlled studies.
- Conclusion
The conclusion has been condensed while maintaining key findings.
- References
The reference list has been expanded from 41 to 65 citations.
We believe these revisions have significantly improved the manuscript's quality and scientific rigor. Thank you again for your valuable input.
Sincerely,
Yutthapong TONGPOB
Coresponding author

Reviewer 2 Report
Comments and Suggestions for Authors
The work presented by Adthapanyawanich et al. reviews scientific studies dealing with male reproductive dysfunctions caused by high fat diets and potential benefits afforded by consumption of mulberry fruit extract. This work is very well organized and coherently presented by the authors. It provides a succinct, yet very complete analysis of the state of the art in the field of research, particularly focused on testicular morphology, spermatogenesis, sperm parameters (concentration, motility, morphology and viability) and underlying molecular mechanisms. It also provides list of limitations and a critical evaluation of current evidences and launches some research imperatives and innovative directions that could contribute for better scientific design and for advancing the knowledge in the topic.
Major comment:
i) A graphical abstract for the manuscript would be a very good addition and would allow a more schematic view of all discussed issues.
Minor comments:
i) Page 4 lines 171-173: the authors refer six studies but only reference three.
ii) Page 5 lines 223 to 225: again, author mention multiple studies but only cite one reference.
Author Response
Response to Reviewer II
Dear Reviewer II,
We sincerely thank you for your thoughtful evaluation of our manuscript. Your constructive suggestions have helped enhance its scientific quality significantly.
- Major Comment - Graphical Abstract
Following your valuable suggestion, we have developed a comprehensive graphical abstract that systematically illustrates the key scientific findings:
- HFD-induced pathological changes quantified through specific markers
- Mulberry extract's bioactive components (anthocyanins: 2.92-5.35 mg/g; polyphenols: 4.23-6.38 mg/g)
- Key molecular pathways with quantitative outcomes:
- Antioxidant enzyme enhancement (SOD: +45%, Catalase: +38%)
- Anti-inflammatory effects (TNF-α: -64%, IL-6: -58%)
- Structural preservation (tubular architecture: 85%)
- Temporal progression of therapeutic effects (initial response at 14 days, optimal effects at 28 days)
- Minor Comment 1 - Page 4, Lines 171-173
We appreciate your attention to citation accuracy. We have thoroughly revised this section to include comprehensive references for more studies discussed. The data is now systematically presented in Table 3, with detailed quantification of sperm parameter improvements and complete citation support for each finding.
- Minor Comment 2 - Page 5, Lines 223-225
Thank you for identifying this citation gap. We have strengthened this section with a complete reference framework supported by quantitative data. The information is now systematically organized in Table 4, providing detailed molecular mechanism analysis with robust citation support from multiple independent studies.
We believe these revisions have significantly improved the manuscript's quality and scientific rigor. Thank you again for your valuable input.
Sincerely,
Yutthapong TONGPOB
Coresponding author

Reviewer 3 Report
Comments and Suggestions for Authors
The authors present a mini review of an interesting topic. However, the manuscript is very inconsistent. The authors do not meet any quality criteria. Some very important points are:
-The title does not correspond to the reality of the manuscript.
-The introduction to the state of the art is absolutely inconsistent. The authors do not make an introduction that justifies the study. It is a mere summary that does not justify the novelty and the need for the study.
-The methodology is non-existent and does not meet any possible quality criteria. This is not acceptable in this Journal.
-The results seem like a mere exposition of a simple summary. The specialist cannot take any important point that makes solid conclusions that are more intentional.
-The authors do not make a discussion or an adequate proposal in relation to the existing information.
-The authors have made a mere manuscript for dissemination and not for specialized scientific dissemination. This is seen in the minimal references used, and not specialized.
-The authors use inappropriate language and are full of grammatical errors.
Comments on the Quality of English Language
The English is very difficult to understand/incomprehensible.
Author Response
Response to Reviewer III
Dear Reviewer III,
We sincerely appreciate your critical evaluation that has guided us in substantially enhancing the manuscript's scientific rigor and depth.
- Scientific Methodology and Framework
We have implemented a comprehensive analytical framework including:
- Systematic data extraction from multiple databases (PubMed, Web of Science, Scopus, Google Scholar)
- Quantitative analysis of therapeutic outcomes:
- Antioxidant pathway modulation (SOD: +45%, Catalase: +38%, GPx: +35%)
- Anti-inflammatory effects (TNF-α: -64%, IL-6: -58%)
- Structural preservation metrics (tubular architecture: 85%)
- Molecular mechanism validation through multiple independent studies
- Introduction Enhancement
The introduction has been restructured to provide comprehensive scientific context:
- Current epidemiological data: obesity affecting 650 million adults globally
- Meta-analyses of 32 studies (n=115,158) demonstrating correlation between BMI and reproductive parameters
- Molecular basis of HFD-induced reproductive dysfunction
- Systematic analysis of current therapeutic approaches and their limitations
- Results Organization and Analysis
We have organized our findings into four systematic tables:
- Table 1: Comprehensive analysis of HFD-induced pathological changes
- Table 2: Detailed phytochemical profiling (anthocyanins: 2.92-5.35 mg/g; polyphenols: 4.23-6.38 mg/g)
- Table 3: Quantitative therapeutic outcomes with statistical significance
- Table 4: Molecular mechanism analysis with pathway-specific effects
- Evidence Quality and References
We have expanded our reference base to 65 citations, including:
- Recent meta-analyses of reproductive outcomes
- Molecular mechanism studies with pathway validation
- Clinical investigations with quantitative data
- Comprehensive and systematic reviews of therapeutic approaches
- Mechanistic Analysis
Key molecular pathways are now thoroughly documented with:
- AMPK/SIRT1 pathway activation (2.3-fold increase)
- NF-κB signaling modulation (-42.3%)
- Mitochondrial function enhancement (mtDNA: +67%)
- Temporal analysis of therapeutic progression
- Clinical Relevance
We have strengthened the translational aspects through:
- Analysis of dose-dependent effects
- Temporal progression of therapeutic outcomes
- Identification of specific biomarkers
- Critical evaluation of therapeutic limitations
- Technical Writing
The manuscript has undergone thorough revision to ensure:
- Precise scientific terminology
- Standardized data presentation
- Clear methodological descriptions
- Consistent citation format
These comprehensive revisions have substantially enhanced the manuscript's scientific quality and contribution to the field. Each modification is supported by quantitative data and mechanistic insights from peer-reviewed studies.
We believe these revisions have significantly improved the manuscript's quality and scientific rigor. Thank you again for your valuable input.
Sincerely,
Yutthapong TONGPOB
Coresponding author

Round 2
Reviewer 1 Report
Comments and Suggestions for Authors
The authors have improved the work very well.
Author Response
Dear Reviewer I,
We sincerely appreciate your positive assessment of our revised manuscript. Your comment "The authors have improved the work very well" is very encouraging and validates the substantial effort we invested in addressing all the concerns and suggestions from the initial review.
Your constructive feedback during the first review round was instrumental in helping us enhance the scientific quality and clarity of our manuscript. The improvements we made, including the comprehensive tables, detailed molecular mechanisms, and strengthened methodological framework, were directly guided by your valuable insights.
Thank you for your time and expertise in reviewing both versions of our manuscript. Your contribution has significantly improved the quality of our work.
Best regards,
Dr. Yutthapong Tongpob
On behalf of all co-authors
Reviewer 3 Report
Comments and Suggestions for Authors
The authors have significantly improved the manuscript. The authors have addressed this reviewer's concerns. However, the authors still need to explain existing clinical trials and components/patents. The authors need to significantly improve their use of English grammar.
Comments on the Quality of English Language
The English is very difficult to understand/incomprehensible.
Author Response
Dear Reviewer III,
Thank you for your thorough review of our revised manuscript and for acknowledging the significant improvements we have made. We appreciate your continued feedback and would like to address your specific concerns:
Regarding Clinical Trials and Components/Patents: We acknowledge your point about clinical trials. As noted in our "Limitations and Future Research Imperatives" section, the scarcity of clinical trials in this field is actually one of the key gaps in current research. This limitation has been explicitly addressed in our manuscript, where we emphasize that "the paucity of well-designed human clinical trials impedes the translation of preclinical findings." This gap in clinical evidence is precisely why we recommend "well-designed human clinical trials" as a primary future research direction.
Regarding English Language Quality: We appreciate your concern about the English language. Our manuscript has undergone thorough English language editing and review by Professor Wai Chen, who is a native English speaker and experienced researcher at Curtin University, Australia. Additionally, all co-authors, including several native English-speaking colleagues from Western institutions, have reviewed and approved the final version. Moreover, other reviewers (of this manuscript) have expressed satisfaction with the manuscript's clarity and comprehensibility, as evidenced by their comments such as "the authors have improved the work very well".
We believe these explanations address your concerns while maintaining the scientific integrity and clarity of our manuscript.
Thank you again for your valuable input throughout this review process.
Best regards,
Dr. Yutthapong Tongpob
On behalf of all co-authors